# Unraveling the Role of miR-200b-3p in Attention-Deficit/Hyperactivity Disorder (ADHD) and Its Therapeutic Potential in Spontaneously Hypertensive Rats (SHR)

**DOI:** 10.3390/biomedicines12010144

**Published:** 2024-01-10

**Authors:** Tung-Ming Chang, Hsiu-Ling Lin, Chih-Chen Tzang, Ju-An Liang, Tsai-Ching Hsu, Bor-Show Tzang

**Affiliations:** 1Pediatric Neurology, Changhua Christian Children’s Hospital, Changhua Christian Hospital, Changhua 500, Taiwan; 128658@cch.org.tw; 2Cardiac Function Examination Room, Chung Shan Medical University Hospital, Taichung 402, Taiwan; echolin01@gmail.com; 3School of Medicine, College of Medicine, National Taiwan University, Taipei City 100, Taiwan; jerrytzang@gmail.com; 4Institute of Medicine, Chung Shan Medical University, Taichung 402, Taiwan; 994103aajj@gmail.com; 5Immunology Research Center, Chung Shan Medical University, Taichung 402, Taiwan; 6Department of Clinical Laboratory, Chung Shan Medical University Hospital, Taichung 402, Taiwan; 7Department of Biochemistry, School of Medicine, Chung Shan Medical University, Taichung 402, Taiwan

**Keywords:** taurine, attention deficit hyperactivity disorder (ADHD), neuropsychiatric disorder, microRNA (miR)-200b-3p, striatum, spontaneously hypertensive rats (SHR), inflammatory factors, spontaneous alternations

## Abstract

Attention-deficit/hyperactivity disorder (ADHD) is a prevalent neurodevelopmental disorder in children with unknown etiology. Impaired learning ability was commonly reported in ADHD patients and has been associated with dopamine uptake in the striatum of an animal model. Another evidence also indicated that micro-RNA (miR)-200b-3p is associated with learning ability in various animal models. However, the association between miR-200b-3p and ADHD–related symptoms remains unclear. Therefore, the current study investigated the role of miR-200b-3p in ADHD-related symptoms such as inattention and striatal inflammatory cytokines. To verify the influence of miR-200b-3p in ADHD-related symptoms, striatal stereotaxic injection of miR-200b-3p antagomir (AT) was performed on spontaneously hypertensive rats (SHR). The antioxidant activity and expressions of miR-200b-3p, slit guidance ligand 2 (Slit2), and inflammatory cytokines in the striatum of SHR were measured using quantitative real-time polymerase chain reaction (RT-qPCR), immunohistochemistry (IHC), immunoblotting, and enzyme-linked immunosorbent assay (ELISA). The spontaneous alternation of SHR was tested using a three-arm Y-shaped maze. The administration of miR-200b-3p AT or taurine significantly decreased striatal tumor necrosis factor (TNF)-α, interleukin (IL)-1β, and IL-6 in SHR, along with increased super-oxide dismutase (SOD) and glutathione peroxidase (GSH-Px) activities and significantly higher spontaneous alternation. In this paper, we show that miR-200b-3p AT and taurine alleviates ADHD-related symptoms in SHR. These findings provide insights into ADHD’s molecular basis and suggest miR-200b-3p as a potential therapeutic target. Concurrently, this study also suggests broad implications for treating neurodevelopmental disorders affecting learning activity such as ADHD.

## 1. Introduction

According to the latest Diagnostic and Statistical Manual of Mental Disorders, fifth edition (DSM-5), attention-deficit/hyperactivity disorder (ADHD) is classified as neurodevelopmental disorders (NDDs). Notably, ADHD is known as a global neuropsychiatric deficit that accounts for approximately 8–12% of all children [1]. Although the etiology of ADHD is complicated and still unclear, neurodevelopmental theory indicates that disruptions in normal brain development during early life can lead to neuropsychiatric symptoms in later life, impacting disorders such as autism spectrum disorder, ADHD, schizophrenia, bipolar disorder, and obsessive compulsive disorder [2]. Additionally, inattentiveness and hyperactivity/impulsiveness are known as the main symptoms of ADHD that lead to various conditions such as depression, epilepsy, and learning deficits [3]. Substantial evidence has reported that gene variants and environmental triggers are the main possible causes of ADHD [4]. Hence, the conventional treatment of ADHD involves a multimodal approach by addressing various aspects of the condition such as cognitive behavioral therapy (CBT), behavioral interventions, exercise, psychoeducation, and medication [1].

MicroRNAs (miRNAs), first discovered in 1993, are known as noncoding RNAs that exhibit pivotal roles in modulating gene expression [5,6]. MiRNAs exist in all animal systems, and some miRNAs are reported to be highly conserved in various species [7,8], which regulate gene expression by binding 3′UTR specific regions of target messenger ribonucleic acid (mRNA) to silence gene expression and promotor-specific regions of target mRNA to induce transcription [9,10]. MiRNAs have gained significant recognition for their roles in neuropsychiatric disorders, with extensive research in recent decades focusing on their impacts on neurodevelopmental conditions like ADHD, autism, and Alzheimer’s disease. These studies primarily explore the influences of miRNAs on cognitive, behavioral, memory, and learning deficits [11,12,13]. Notably, a recent study indicated that the miRNA expression profile, including miR-4516, miR-6090, miR-4763-3p, miR-4281, and miR-4466, has great diagnostic accuracy and specificity in assessing ADHD [14]. A similar result was also reported, where the expression levels of miR-126-5p, miR-140-3p, and miR-30e-5p in total white blood cells (WBCs) revealed great clinical potential as diagnostic and therapeutic biomarkers for ADHD [15]. Although these miRNAs change significantly in ADHD patients as compared to healthy individuals, the precise roles of each miRNA and their interactions in the development of ADHD are still unclear. However, the application of miRNAs aids in pinpointing potential targets for therapy and paves the way for the possibility of genetic mRNA treatments.

Taurine, known as a free β-amino acid, is a very abundant neurotransmitter in the human nervous system that exhibits diverse physiological roles such as a regulator of calcium transport and homeostasis, an osmolyte, and a trophic factor in the development of central nervous system [16,17,18,19]. Taurine has been demonstrated to have therapeutic potential in a broad range of disorders, including neurodegenerative diseases such as Alzheimer’s disease, Parkinson’s disease, epilepsy [20], muscle atrophy [21], congestive heart failure [22,23], rheumatoid arthritis [24], thrombosis [25], and lipid metabolism disorders [26]. Additionally, taurine exhibits neuroprotective properties through its ability to stabilize cell membranes, inhibits reactive oxygen species (ROS)-producing enzymes, and indirectly acts as an antioxidant by maintaining cellular redox balance, effectively safeguarding neuronal health [27]. Recent evidence revealed that amino acid supplementation may influence genetic expression through epigenetic mechanisms such as DNA hypermethylation, potentially altering the outcomes of NDDs [28]. Maternal protein restriction during gestation in spontaneously hypertensive rats (SHR), a well-documented ADHD animal model for investigating the treatment of ADHD [29], led to a positive correlation between DNA hypermethylation at the CpG island of the renal prostaglandin E receptor 1 (Ptger1) gene and increased Ptger1 mRNA expression in offsprings. Interestingly, the study also found that post-weaning dietary adjustments, either to a low-protein or high-protein diet, could modify the Ptger1 DNA hypermethylation caused by fetal malnutrition [28]. These findings revealed that nutritional supplements like taurine may influence genetic expressions through epigenetic mechanisms like DNA hypermethylation, potentially altering the outcomes of NDDs.

Compared with the controls, SHR receiving high-dose taurine (45 mmol/kg) for four weeks revealed significantly decreased hyperactive behavior by reducing inflammatory cytokines, functional connectivity (FC) signal, and the mean amplitude of low-frequency fluctuation (mALFF) in the bilateral hippocampus [30]. Additionally, the administration of high-dose taurine reduced the dopamine uptake in striatal synaptosomes of SHR and increased the spontaneous alternation of SHR [31]. Although these findings suggested the ameliorating effects of taurine on ADHD-like symptoms in SHR, the mechanism of taurine in improving the symptoms of ADHD is still unclear. Adding to this, recent studies have linked the expression of miR-200b-3p with learning ability in various disease models [32,33]. Building on this evidence, the current study explores the role of miR-200b-3p in SHR, aiming to evaluate its therapeutic potential for ADHD. Therefore, the current study investigated the role of miR-200b-3p in SHR to assess its therapeutic potential in ADHD. We hypothesized that the administration of either taurine or miR-200b-3p antagomir (AT) positively influences the neurobiological and behavioral symptoms associated with ADHD in the SHR. Specifically, taurine or miR-200b-3p AT will lead to a reduction in the miR-200b-3p level and an increase in its target, Slit2, expression in the striatum of SHR, accompanied by a reduction in oxidative stress in the striatum and improved inattention behavior.

## 2. Materials and Methods

This study was designed based on previous publications [31,34] and explored the effects of taurine and miR-200b-3p AT on ADHD-related symptoms using spontaneously hypertensive rats (SHR), a model for ADHD. The research involved dividing rats into groups for various treatments and control conditions, administering taurine diets, stereotaxic injections of miR-200b-3p antagomir, and conducting behavioral, molecular, and biochemical analyses. Techniques such as RT-qPCR, ELISA, immunohistochemistry, and immunoblotting were used to assess mRNA expression, inflammatory cytokines, antioxidant enzyme levels, and protein expression as described elsewhere [35,36,37]. Additionally, the study evaluated working memory using a Y-maze test and employed statistical analysis to interpret the data [38].

### 2.1. Animals and Experimental Procedure

To investigate the influence of taurine and miR-200b-3p antagomir on ADHD-related symptoms, the spontaneously hypertensive rat (SHR/NCrlCrlj; SHR), a valid ADHD animal model, and the Wistar Kyoto rat (WKY/NCrlCrlj; WKY), control rats for SHR, were adopted in this study [34]. All rats were obtained at three weeks old from the National Laboratory Animal Center, Taipei, Taiwan, and separated into five groups (five rats/group), including the Control, Taurine, Sham, miR ATNC (miR-200b-3p antagomir negative control), and miR AT (miR-200b-3p antagomir) groups. The animals were kept in a facility at 22 ± 2 °C with a 12/12 h light–dark cycle. Experimental handling was approved and supervised by the Institutional Animal Care and Use Committee at Chung Shan Medical University (IACUC approval number: 2136). The taurine dose used in this study was 45 mmol/kg diet according to our previous publication [30,31]. At four weeks of age, rats from the taurine group were administered a taurine diet, while those rats from the other groups were fed a standard chow diet until they reached eight weeks of age. Stereotaxic injection surgery was performed on rats from the Sham, miR ATNC, and miR AT groups at five weeks of age. The Y-maze test was conducted for all rats one day prior to their sacrifice by CO_2_ asphyxiation at eight weeks of age. The striatum tissue of rats from each group was resected and kept in a −80 °C freezer before analysis.

### 2.2. MicroRNA and Striatal Stereotaxic Injection

To assess the impacts of blocking striatum miR-200b-3p expression in SHR, rat miR-200b-3p antagomir (AT) and the antagomir-negative control (ATNC) were purchased (BioLion Technology Co., Ltd., Taipei, Taiwan). Five nmol miR-200b-3p AT and ATNC in 1 μL PBS were mixed with 1 μL HiPerFect transfection reagent (Cat. #: 301705, Qiagen, Germantown, MA, USA) prior to injection into the striatum of SHR. The striatal stereo-taxic injection was performed as described elsewhere [35]. Briefly, SHR were intraperitoneally injected with urethane (1.25 g/kg) to anesthetize them, and they were placed on an animal heating pad. Next, the rats were fixed in a stereotactic apparatus, and a hole was drilled in the skull. The mixed solution was then injected (1 μL/min) into the left striatum of rats using a 10 μL Hamilton syringe (Sigma-Aldrich, St. Louis, MO, USA) connected to a microinfusion pump (Stoelting Co., Wood Dale, IL, USA). The skin was sutured after injection.

### 2.3. Quantitative Real-Time PCR (RT-qPCR)

To detect the mRNA expression in the striatum of rats, RT-qPCR analysis was performed based on a previous study [33]. Total RNA was extracted from the left striatum of rats using miRNeasy Kits (Cat. #: 217604, Qiagen, Germantown, MA, USA) and subsequently reversed to complementary DNA (cDNA) using miRCURY LNA RT Kit (Cat. #: 339340, Qiagen, Germantown, MA, USA). Quantitative real-time PCR (RT-qPCR) was completed using miRCURY LNA SYBR^®^ Green PCR Kits (Cat. #: 339345, Qiagen, Germantown, MA, USA) and analyzed using Applied Biosystems StepOne Plus Real-Time PCR System. The specific primers used in this study are shown in Table 1. The relative gene expression was analyzed as described elsewhere [36].

### 2.4. Enzyme-Linked Immunosorbent Assay (ELISA)

To detect the levels of inflammatory cytokines and the activity of antioxidant enzymes, ELISA tests were performed. The striatum tissues were homogenized, and the supernatants were collected after centrifugation. The levels of superoxide dismutase (SOD) and glutathione peroxidase (GSH-Px) in rat striatum were measured using ELISA kits purchased from MyBioSource (Cat. #: MBS266897, MyBioSource, San Diego, CA, USA; Cat. #: MBS032696, MyBioSource, San Diego, CA, USA). The contents of tumor necrosis factor-α (TNF-α), interleukin-1β (IL-1β), and interleukin-6 (IL-6) were also measured using ELISA kits obtained from Invitrogen (Cat. #: KRC3011, Invitrogen, Thermo Fisher Scientific, Waltham, MA, USA; Cat. #: BMS630 Invitrogen, Thermo Fisher Scientific, Waltham, MA, USA; Cat. #: ERA31RB, Invitrogen, Thermo Fisher Scientific, Waltham, MA, USA).

### 2.5. Immunohistochemistry (IHC)

To detect the expression of the Slit2 protein in the striatum of SHR, immunohistochemistry (IHC) was performed as described elsewhere [37]. Animals were euthanized by carbon dioxide. The striatum tissues were excised, soaked in 10% formalin, and subsequently embedded with paraffin wax. The embedded tissues were sectioned into 5 μm slices and incubated overnight with antibodies against rat Slit2 (Cat. #: ab7665, Abcam, Waltham, MA, USA). Finally, the sections were observed and quantified using the automated Tissue-FAXS PLUS system (TISSUE GNOSTICS, Vienna, Austria).

### 2.6. Immunoblotting

To detect the Slit2 protein expression in the striatum tissue of SHR with different treatments, immunoblot was conducted as described in our previous study [31]. Total proteins were extracted from the striatal tissues in PRO-PREP™ buffer (iNtRON Bio-technology, Inc., Seongnam, Republic of Korea), and the concentrations of protein were measured according to a modified Bradford’s assay using a spectrophotometer (Hitachi U3000, Tokyo, Japan) at 595 nm. The proteins were separated into a sodium dodecyl sulfate polyacrylamide gel electrophoresis (SDS-PAGE) via electrophoresis and then transferred onto a nitrocellulose membrane (Amersham Biosciences, Piscataway, NJ, USA). After blocking the membrane with 5% nonfat dry milk, antibodies against rat Slit2 (Cat. #:ab7665, Abcam, Waltham, MA, USA), or β-actin (Cat. #: MAB1501, Merck Millipore, Burlington, MA, USA), they were incubated for 2 h with mild shaking. Subsequent incubation of horseradish peroxidase (HRP) conjugated secondary antibody for another hour was performed. For detecting the antigen–antibody complexes, Immobilon Western Chemiluminescent HRP Substrate (Millipore, Burlington, MA, USA) and an imaging analyzer (GE ImageQuant TL 8.1, GE Healthcare Life Sciences, Pittsburgh, PA, USA) were used.

### 2.7. Spontaneous Alternation

The working memory of the rats was assessed according to a previous method described elsewhere [31,38]. Briefly, a three-arm Y-shaped maze with 200 lx illumination was used to test the spontaneous alternation. The three arms are angled 120° to each other, and each arm is 20 inches long, 4 inches wide, and 15 inches high. A rat is considered to have entered the arm when its four paws are in the arm. Spontaneous alternation was defined as the entry of all three arms in consecutive choices in triplet sets overlapped, and the percentage of spontaneous alternation was shown as (actual alternations/maximal alternations) ×100. The maximum number of alternations was defined as the total number of arm entries minus two.

### 2.8. Statistical Analysis

GraphPad Prism 5.0 software was used to analyze the experimental data. The data were presented as mean ± S.D. Two-way ANOVA with Bonferroni’s post hoc test for multiple comparisons was used to analyze the effects of rat type and treatment, as well as the interaction of these two factors. One-way ANOVA with Tukey’s multiple comparisons post hoc test was performed to determine the significance of different treatments of SHR. A *p*-value less than 0.05 (*p* < 0.05) was considered as statistically significant.

## 3. Results

### 3.1. Effect of High-Dose Taurine on Expressions of miR-200b-3p and Silt2 in the Striatum of WKY and SHR

We first employed RT-qPCR and immunoblotting to assess the expression levels of miR-200b-3p and Slit2 in the striatum of both WKY and SHR. As shown in Figure 1A, the expression of miR-200b-3p was significantly higher in the striatum of SHR as compared to WKY rats. No significant difference in miR-200b-3p expression was observed between the WKY rats administered high-dose taurine and those on a control diet. A significantly decreased miR-200b-3p level was detected in SHR treated with high-dose taurine compared to the SHR controls (Figure 1A). No significant difference in the expressions of Slit2 mRNA, a target gene of miR-200b-3p, and Slit2 protein was observed in the striatum of WKY rats treated with high-dose taurine (Figure 1B,C). Notably, significantly upregulated levels of both Slit2 mRNA and Slit2 protein were detected in the striatum of SHR treated with high-dose taurine compared to the SHR controls (Figure 1B,C).

### 3.2. Effects of miR-200b-3p Antagomir on miR-200b-3p and Slit2 Protein Expressions in Striatum of SHR

The expressions of miR-200b-3p and Slit2 protein in the striatum of SHR were detected using RT-qPCR and immunoblotting analysis, respectively. Compared with the controls, the level of miR-200b-3p was significantly decreased in the striatum of SHR treated with high-dose taurine as well as those rats treated with miR-200b-3p antagomir (Figure 2A). Additionally, a significantly increased expression of Slit2 protein was detected in the striatum of SHR treated with high-dose taurine and miR-200b-3p antagomir, respectively (Figure 2B). Immunohistochemistry (IHC) was also performed to confirm the expressions of the Slit2 protein in the striatum of SHR with different treatments. A significantly higher Slit2 protein level was detected in the striatum of SHR treated with high-dose taurine and miR-200b-3p antagomir, respectively, compared with the controls (Figure 3A,B).

### 3.3. MiR-200b-3p Antagomir Attenuated the Expressions of Inflammatory Cytokines and Increased the Activity of Antioxidant Enzymes

To verify the effects of miR-200b-3p antagomir on inflammation-related factors, the expressions of TNF-α, IL-1β, and IL-6 in the striatum of SHR were measured. Significantly lower levels of TNF-α, IL-1β, and IL-6 mRNA and their protein expressions were observed in the striatum of SHR treated with high-dose taurine and miR-200b-3p antagomir, respectively, compared to the controls (Figure 4A,B). Additionally, a significantly higher activity of GSH-Px and SOD was detected in the striatum of SHR treated with high-dose taurine and miR-200b-3p antagomir, respectively (Figure 5A,B).

### 3.4. MiR-200b-3p Antagomir Improves Working Memory in SHR

To verify the effects of miR-200b-3p antagomir on working memory in SHR, arm entries, and spontaneous alternation tests were performed with a three arms Y-maze test (Figure 6A). A significantly lower total number of arm entries was observed in SHR treated with high-dose taurine compared to those of the control group. A similar result was also detected in SHR treated with miR-200b-3p antagomir as compared with those from the Sham and miR ATNC groups, respectively (Figure 6B). Additionally, a significantly higher percentage of spontaneous alternation was detected in SHR treated with high-dose taurine and miR-200b-3p antagomir compared to the control groups (Figure 6C).

In summary, our experimental results revealed that the administration of taurine or miR-200b-3p AT significantly ameliorates the striatal proinflammatory cytokines, including TNF-α, IL-1β, and IL-6, and increases the activity of anti-oxidant enzyme activity, including SOD and GSH-Px, in SHR. Concurrently, a significant increase in spontaneous alternation was detected in SHR treated with taurine or miR-200b-3p AT.

## 4. Discussion

MicroRNAs (miRNAs) are known as a family of untranslated single-stranded RNAs with approximately 22 nucleotides in length [39]. Although the detailed mechanism of miRNAs is still not fully understood, the function of most miRNAs in mammals is thought to inhibit the target gene translation by mRNA degradation, which plays an essential role in controlling cell division, differentiation, and death [40]. In recent decades, miRNAs have been versatile, being used in the diagnosis, prognosis, and as therapeutic targets in many diseases, including cancers, CNS disorders, hepatic diseases, autoimmune disorders, and cardiovascular diseases [41,42,43,44,45]. Although increasing studies have been focused on investigating the miRNAs in attention-deficit/hyperactivity disorder (ADHD) [15], the roles and applications of miRNAs in ADHD are still very limited. For the first time, we reported that high-dose taurine significantly attenuated the level of miR-200b-3p along with increased Slit2 protein in the striatum of SHR, leading to attenuated expressions of inflammatory cytokines, elevated activity of antioxidants, and increased spontaneous alternations. These findings indicated the involvement and regulatory roles of miR-200b-3p in ADHD-like symptoms and suggested miR-200b-3p as a therapeutic target for ADHD-like symptoms.

MiR-200b-3p is a member of miR-200b family, which contains miR-200a, miR-200b, miR-200c, miR-429, and miR-141. Although most studies investigating miR-200b-3p are related to its regulation and mechanism on malignant phenotype tumors [46], miR-200b-3p also exhibits modulatory roles in many physiological and pathological processes, including the formation of insulin-producing cells [47], fetal cartilage differentiation [48], preeclampsia [49], wound healing [50], and neuropathological disorders [51]. However, very limited information is known about the roles of miR-200b-3p in ADHD-like symptoms. Recently, upregulated miR-200b-3p was reported to be associated with the development of brain arteriovenous malformations [52]. Another study also indicated that miR-200b-3p antagomir improved spatial and learning memory loss in hypoxia-ischemia animals [53]. These findings indicated that the decline of miR-200b-3p expression may reveal a protective effect on brain development as well as improved spatial and learning memory, which may provide a possible explanation for the effects of the downregulated miR-200b-3p level in the striatum of SHR treated with high-dose taurine. However, more investigations are still required to verify the precise mechanism of miR-200b-3p in the pathological processes of ADHD and its related symptoms.

The current study reported the decreased expressions of proinflammatory cytokines and the increased activity of antioxidant enzymes in the striatum of SHR receiving high-dose taurine. However, information about the role of miR-200b-3p on the expression of proinflammatory cytokines and antioxidant enzyme activity is still unclear. Notably, the enhancement of various inflammatory cytokines by miR-200b-3p was reported in an avian model [54]. A recent study indicated that upregulation of gga-miR-200b-3p promotes macrophage differentiation and enhances the expressions of proinflammatory cytokines such as TNF-α, IL-1β, IL-6, and IL-12 by directly targeting monocyte to macrophage differentiation-associated (MMD) [54]. This finding indicated evidence that miR-200b-3p directly regulates the expressions of various proinflammatory cytokines. Although no direct evidence indicated the act of miR-200b-3p on antioxidant enzymes such as superoxide dismutase (SOD) and glutathione peroxidases (GPxs), a recent review study indicated the association between ROS and the miR-200 family [55]. Notably, compelling evidence has indicated the existence of a reciprocal connection between antioxidant enzyme activity and the miR-200b family that maintains the cellular redox balance [56]. Apart from the findings mentioned above, the downregulated inflammatory cytokines and upregulated antioxidant activity may also be caused by the action of taurine [57], which also provides another rationale for the findings in this study.

In this study, SHR fed with high-dose taurine revealed significantly decreased miR-200b-3p in the striatum. In fact, current research on taurine-regulated miRs and their related mechanisms is still very limited. Therefore, very little information is known about the regulatory mechanism of taurine on miR-200b-3p expression. Interestingly, in an ex vivo study of adaptive osmotic response under hypertonic stress, significantly upregulated Na+/Cl−-taurine transporter, a hypertonic responsive gene, was due to the downregulated levels of miR-29b-3p and miR-200b-3p [58]. As taurine and the taurine transporter are known to play essential roles in the modulation of neuron osmosis and neurotransmitter balance [19], this finding may provide a possible explanation for the mechanism of the regulatory role of taurine on the miR-200b-3p level. Further investigations are required to verify the detailed network of how taurine downregulates the level of miR-200b-3p.

Although there are various animal models for investigating ADHD, SHR are currently recognized as the most appropriate animal model for ADHD. Various studies have reported that attention-deficit/hyperactivity disorder (ADHD) is linked to changes in encoding processes, specifically in working or short-term memory [59]. Interestingly, the spontaneously hypertensive rats (SHR) displays certain dysfunctional domains associated with ADHD [60,61]. Indeed, spontaneously hypertensive rats (SHR) exhibit symptoms related to hypertension [62] and ADHD-like syndromes such as inattention, hyperactivity, and impulsivity [60]. Moreover, the dysregulation of dopamine signaling between the frontal cortex and the striatum is known as an important occurrence associated with behavioral changes in ADHD [63]. Notably, similar deficits in energy metabolism, dopaminergic signaling, and neural development are also reported in the striatum of SHR [31,64]. Therefore, in this study, SHR were the appropriate animal model to investigate the effects of taurine and miR-200b-3p AT on ADHD-related symptoms.

In order to assess clinical relevance, it is essential to compare the doses administered to animals in this study with those necessary for humans. Previous evidence has suggested that the taurine intake from daily food consumption is approximately 58 mg [65], aligning with the taurine concentration (30 to 160 mg) found in a standard 100 g taurine-rich food like fish, beef, or pork [66]. However, high-dose taurine has been demonstrated as nontoxic to humans [67] and has been applied in different pathophysiological conditions such as skeletal muscle disorders and heart failure [21,68]. For the clinical treatment of congestive heart failure [69], hypertension [70], and dystrophic myotonia [71], taurine is used at doses as high as 6 g per day or more. Notably, the highest tolerable dose of taurine was identified as 21 g per day in a clinical trial aimed at managing epilepsy [72]. These findings suggest that high-dose taurine intake is well tolerated for the treatment of various human pathological conditions. The taurine dose used in this study was 45 mmol taurine/kg diet (5.6 g taurine/kg diet), which is equivalent to a dose of 0.9 g taurine/kg diet in humans [73]. The dose of taurine used in this study is much lower than that used for various diseases mentioned above [72], which provides rational support for ADHD treatment.

Certain concerns within this study require further emphasis. First, there are some issues that need to be raised in the animal behavior experiment of this study. Since only the striatum was measured, this study is still limited in interpreting the experimental results of taurine and miR-200b-3p antagomir affecting animal behavior. Additionally, the test of spontaneous alternation alone may not provide a comprehensive assessment of ADHD symptoms or working memory. Therefore, other tests such as a locomotion test, Morris water maze test, open field test, and Barnes maze test may be merited to clarify the effects of taurine and miR-200b-3p antagomir on learning and cognition in SHR in the future [74,75]. Additionally, this study shows that the elevated miR-200b-3p levels in the striatum of SHR were reduced by the administration of taurine, resulting in improved attention. However, it is worth noting that taurine is known to influence a broad range of miRNAs involved in various physiological and pathological processes, including CNS development, hormone metabolism, inflammation, and cognitive function [76,77,78,79,80]. Therefore, further investigations are warranted to better understand the specific regulatory network of miRNAs modulated by taurine, particularly in the context of improving ADHD-like symptoms in SHR. Moreover, it is important to acknowledge that microRNA-based therapy faces several limitations and challenges that must be addressed before translating these findings into clinical applications. Notably, issues such as immunotoxic reactions and suboptimal delivery systems have been identified as significant hurdles in recent research [81,82]. Hence, the development of miRNA therapies with low toxicity, high effectiveness, and precise targeting is essential for advancing miRNA-based drug development.

## 5. Conclusions

Irregular dopamine signaling between the frontal cortex and the striatum is recognized as a noteworthy phenomenon associated with behavioral changes in ADHD [63,83]. Therefore, in this study, we investigated the effects of taurine supplementation on the striatum of SHR. As shown in Figure 7, SHR fed with taurine exhibited a noteworthy reduction in miR-200b-3p expression in the striatum of the brain, accompanied by diminished expressions of inflammatory cytokines, including TNF-α, IL-1β, and IL-6, and heightened antioxidant enzyme activity, including SOD and GSH-Px. Intriguingly, SHR treated with the miR-200b-3p antagomir also displayed reduced expressions of inflammation-related cytokines and increased antioxidant enzyme activity in the striatum of the brain. Furthermore, regardless of whether the SHR were administered taurine or injected with the miR-200b-3p antagomir, a significant improvement in their working memory was observed. These findings suggest that the miR-200b-3p antagomir reveals a similar function to taurine and highlights its potential as a therapeutic target for ADHD treatment.

## Figures and Tables

**Figure 1 biomedicines-12-00144-f001:**
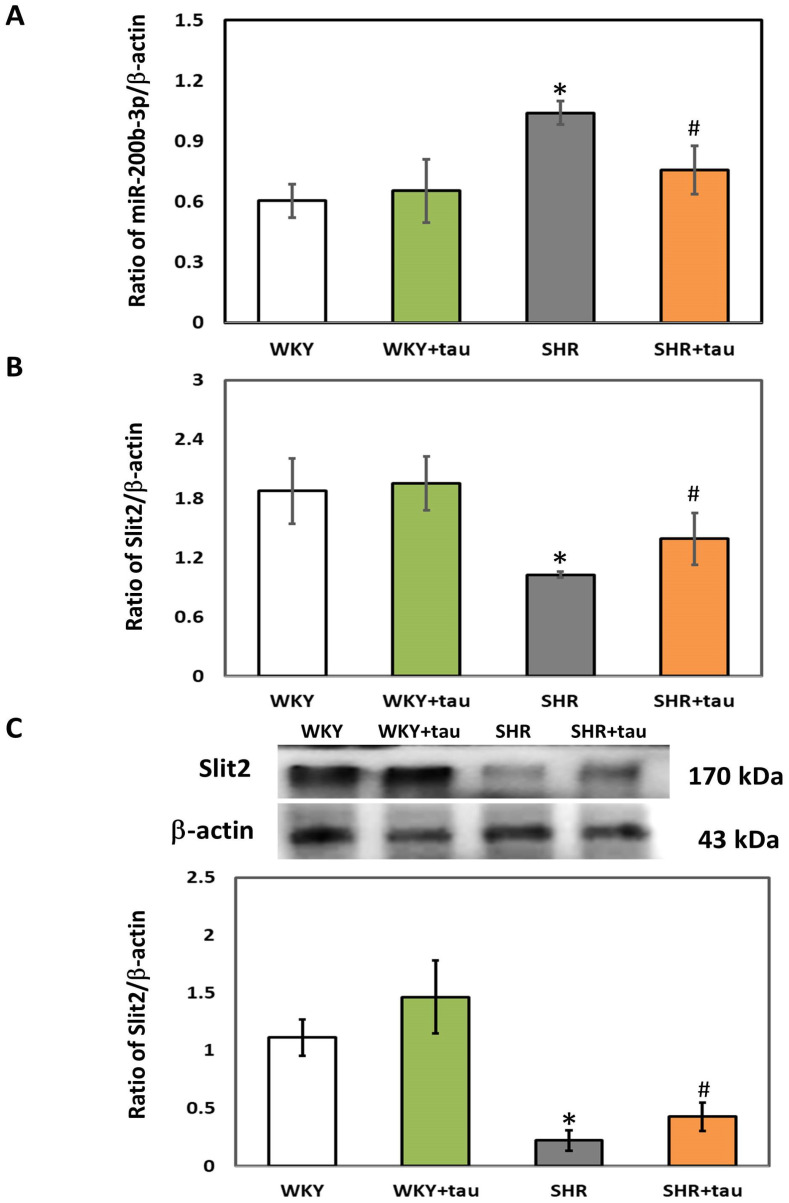
Comparison of miR-200b-3p and Slit2 expressions in the striatum of WKY and SHR. (**A**) miR-200b-3p, (**B**) Slit2 mRNA, and (**C**) Slit2 protein in the striatum of WKY and SHR from different groups (*n* = 5 per group). Data are shown as mean ± S.D. The symbols, * *p* < 0.05 and # *p* < 0.05, indicate significant differences compared with the WKY group and SHR group, respectively, using two-way ANOVA with Bonferroni’s post hoc test. WKY (fed with Cho diet); WKY + tau (fed with 45 mM taurine); SHR (fed with Cho diet); SHR + tau (fed with 45 mM taurine).

**Figure 2 biomedicines-12-00144-f002:**
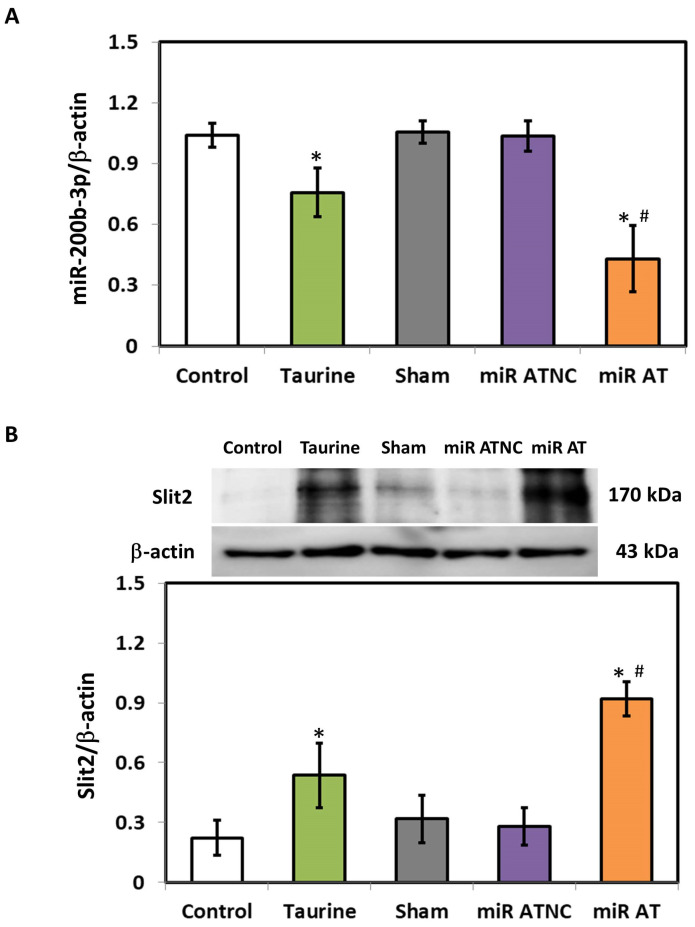
Expression of miR-200b-3p and Slit2 protein in SHR treated with miR-200b-3p antagomir. (**A**) miR-200b-3p and (**B**) Slit2 protein in the striatum of SHR from different groups (*n* = 5 per group). Data are shown as mean ± S.D. The symbols, * *p* < 0.05, and # *p* < 0.05, indicate significant differences compared with the Control group and Sham group, respectively, using one-way ANOVA with Tukey’s multiple comparisons post hoc test. Control (fed with Cho diet); Taurine (fed with 45 mM taurine); Sham (fed with Cho diet); miR ATNC (injection of miR-200b-3p antagomir negative-control); miR AT (injection of miR-200b-3p antagomir).

**Figure 3 biomedicines-12-00144-f003:**
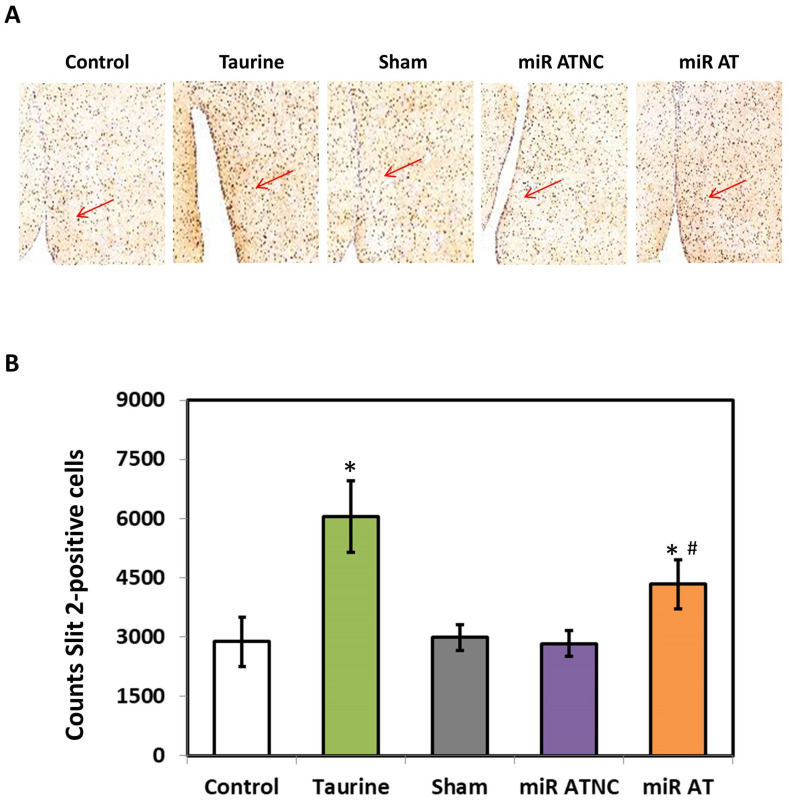
Immunohistological stainings for Slit2 proteins. (**A**) Representative images of the striatal section with immunohistological stainings of SHR with different treatments (*n* = 5 per group). The arrow indicates the expression of Slit2 proteins. (**B**) Quantified results for Slit2 protein expression. The symbol, * *p* < 0.05, and # *p* < 0.05, indicate significant differences compared with the Control group and Sham group, respectively, using one-way ANOVA with Tukey’s multiple comparisons post hoc test. Control (fed with Cho diet); Taurine (fed with 45 mM taurine); Sham (fed with Cho diet); miR ATNC (injection of miR-200b-3p antagomir negative-control); miR AT (injection of miR-200b-3p antagomir).

**Figure 4 biomedicines-12-00144-f004:**
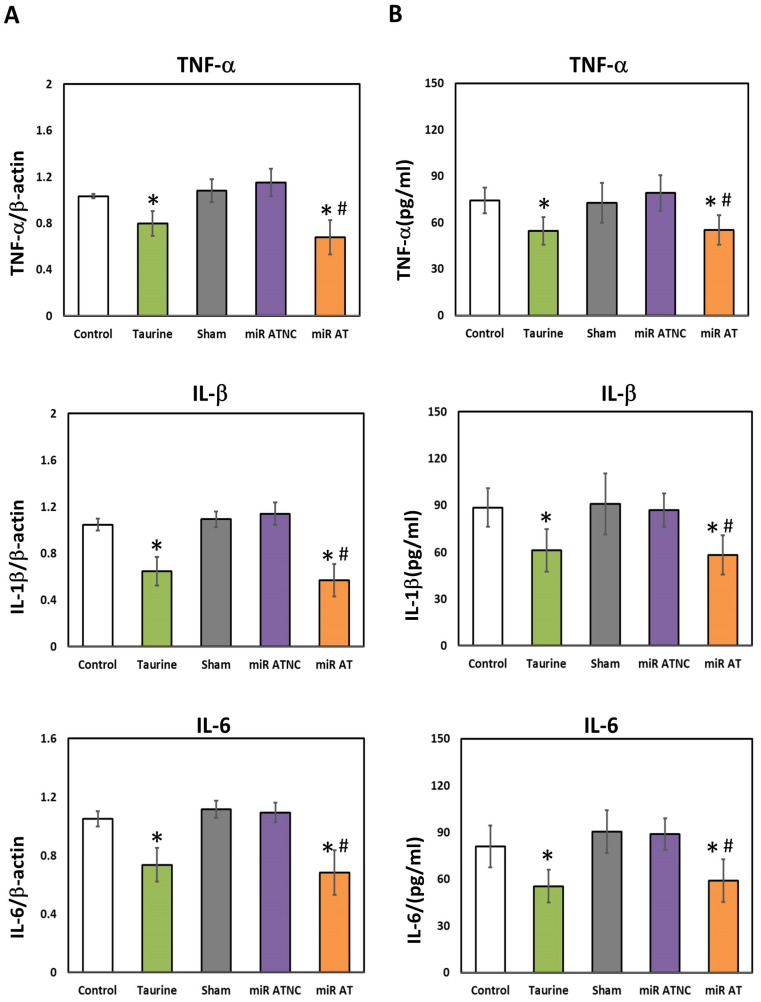
Levels of inflammatory cytokines in the striatum of SHR treated with miR-200b-3p antagomir. (**A**) Relative mRNA expression of TNF-α, IL-1β, and IL-6 and (**B**) concentrations of TNF-α, IL-1β, and IL-6 in the striatum of SHR from different groups (*n* = 5 per group). Data are shown as mean ± S.D. The symbol, * *p* < 0.05, and # *p* < 0.05, indicate significant differences compared with the Control group and Sham group, respectively, using one-way ANOVA with Tukey’s multiple comparisons post hoc test. Control (fed with Cho diet); Taurine (fed with 45 mM taurine); Sham (fed with Cho diet); miR ATNC (injection of miR-200b-3p antagomir negative-control); miR AT (injection of miR-200b-3p antagomir).

**Figure 5 biomedicines-12-00144-f005:**
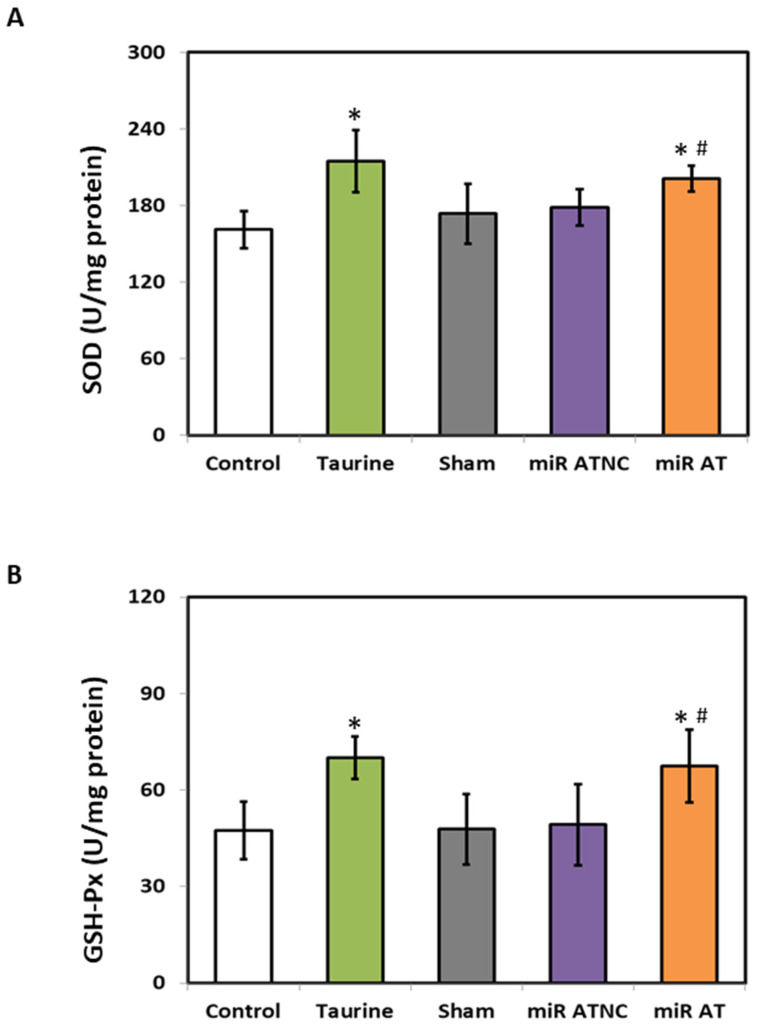
The levels of SOD and GSH-Px in the striatum of rats treated with miR-200b-3p antagomir. The activity of (**A**) SOD and (**B**) GSH-Px in the striatum of SHR from different groups (*n* = 5 per group). Data are shown as mean ± S.D. The symbol, * *p* < 0.05, and # *p* < 0.05, indicate significant differences compared with the Control group and Sham group, respectively, using one-way ANOVA with Tukey’s multiple comparisons post hoc test. Control (fed with Cho diet); Taurine (fed with 45 mM taurine); Sham (fed with Cho diet); miR ATNC (injection of miR-200b-3p antagomir negative-control); miR AT (injection of miR-200b-3p antagomir).

**Figure 6 biomedicines-12-00144-f006:**
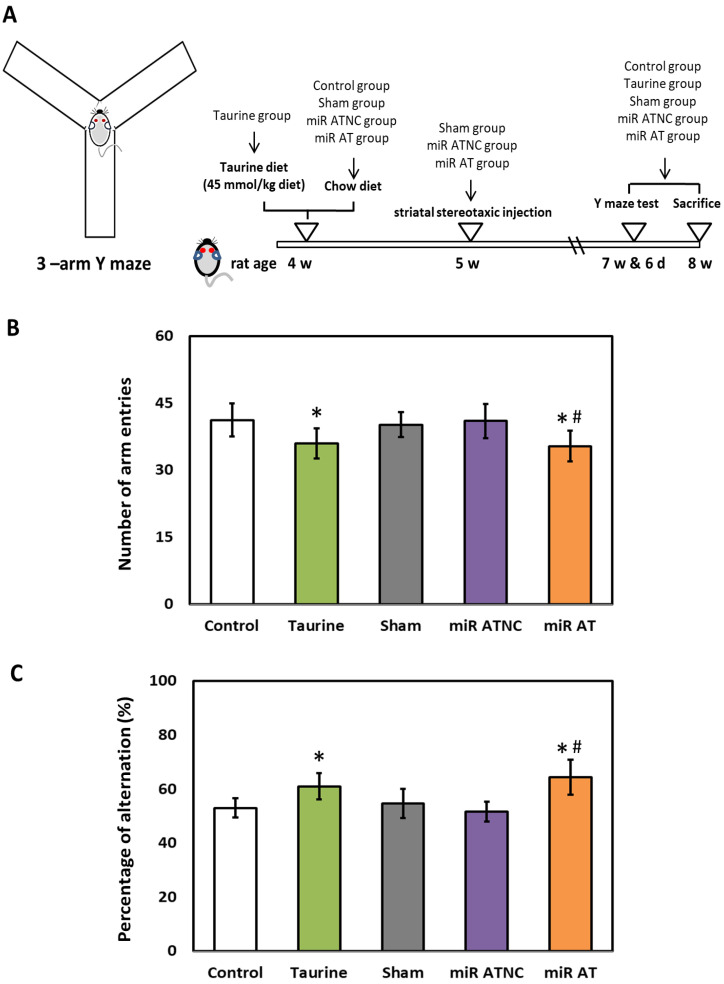
Effects of miR-200b-3p antagomir on inattention of SHR. (**A**) Schematic diagram of 3-arm Y-maze device and experimental design. (**B**) Total arm entries and (**C**) spontaneous alternation behavior in SHR from different groups (*n* = 5 per group). Data are shown as mean ± S.D. The symbol, * *p* < 0.05, and # *p* < 0.05, indicate significant differences compared with the Control group and Sham group, respectively, using one-way ANOVA with Tukey’s multiple comparisons post hoc test. Control (fed with Cho diet); Taurine (fed with 45 mM taurine); Sham (fed with Cho diet); miR ATNC (injection of miR-200b-3p antagomir negative-control); miR AT (injection of miR-200b-3p antagomir).

**Figure 7 biomedicines-12-00144-f007:**
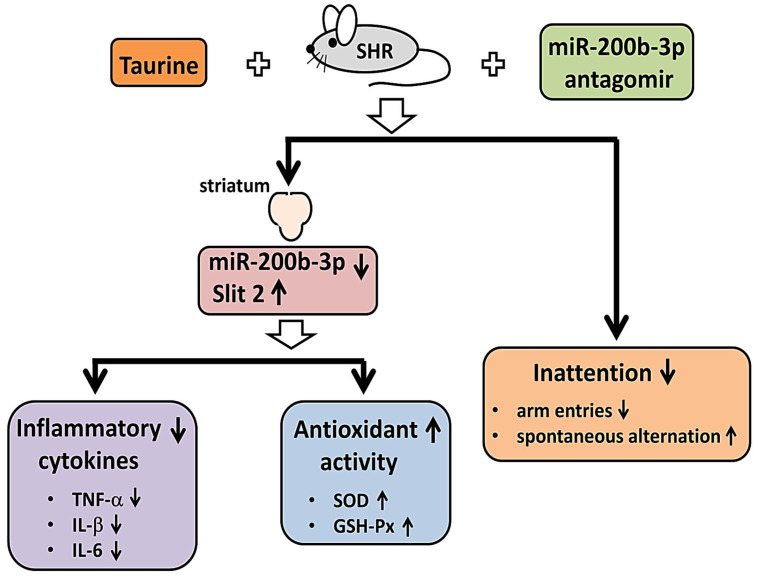
Graphical abstract of the effects of both taurine and miR-200b-3p antagomir on striatum of SHR and inattention.

**Table 1 biomedicines-12-00144-t001:** Primers for quantitative real-time PCR (qRT-PCR).

Gene	Forward Primer	Reverse Primer
*Slit2*	5′-CGCCAAAGGGATTCAAGTGT-3′	5′-CACTGGCATATTGGTTCATTCA-3′
*β-actin*	5′-CCCATCTATGAGGGTTACGC-3′	5′-TTTAATGTCACGCACGATTTC-3′
*TNF-α*	5′-TCAGCCGATTTGCCATTTCAT-3′	5′-ACACGCCAGTCGCTTCACAGA-3′
*IL-1β*	5′-GTCCTTTCACTTGCCCTCAT-3′	5′-CAAACTGGTCACAGCTTTCGA-3′
*IL-6*	5′-AATGCCTCGTGCTGTCTGACC-3′	5′-GGTGGGTGTGCCGTCTTTCATC-3′

*Slit2*: Slit guidance ligand 2; *TNF*: Tumor necrosis factor; *IL*: Interleukin.

## Data Availability

The data presented in this study are available upon request from the corresponding author.

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
