# Peer review of "Unraveling the Role of miR-200b-3p in Attention-Deficit/Hyperactivity Disorder (ADHD) and Its Therapeutic Potential in Spontaneously Hypertensive Rats (SHR)"

_biomedicines, 2024, doi:10.3390/biomedicines12010144_

Round 1
Reviewer 1 Report
Comments and Suggestions for Authors
Dear Author,
I have carefully reviewed your submitted research paper entitled "Taurine ameliorates ADHD-like symptoms in SHR rats is associated with miR-200b-3p". During the review process, I have noticed several areas that require improvement and supplementation. Here are my review comments:
Please display the error bars completely in the figures:
In your submitted manuscript, some of the figures have incomplete error bars. To ensure the accuracy and reliability of the data, I suggest displaying the error bars in their entirety so that readers can have a clearer understanding of the variability of the results. Please ensure that the length of the error bars encompasses the range of variation for all data points.
Please provide the complete raw data:
To further validate and reproduce your research findings, I recommend providing the complete raw data. This will assist other researchers in verifying your study and enhance the reproducibility and reliability of your research. Please ensure the accuracy and completeness of the raw data, and clearly indicate in the manuscript how the data were obtained and processed.
Please include a picture of the apparatus used in the Spontaneous alternation experiment:
In your study, you mentioned the Spontaneous alternation experiment but did not provide a picture of the apparatus used. The apparatus image is crucial for readers to understand the experimental design and procedures. I suggest including a photograph of the Spontaneous alternation experiment apparatus and providing a description in the manuscript to help readers better understand the specific details of the experiment.
The Western blotting (WB) bands in Figure 2B are not aesthetically pleasing:
I noticed some aesthetic issues with the WB bands in Figure 2B. To enhance the credibility and visual impact of your research findings, I recommend optimizing the presentation of the WB experiment results to ensure the clarity and aesthetics of the bands. You may consider using higher-resolution images, adjusting exposure times, or optimizing image processing methods to improve the image quality.
These are my review comments on your submitted manuscript. I hope these suggestions will be helpful for you to further refine and improve your research. If you need any further clarification or have any other questions, please feel free to contact me.
Thank you for your efforts and contributions.
Best regards,
Reviewer
Comments on the Quality of English LanguageThe writing level of the provided abstract is fairly good. The sentences are generally clear and concise, and the information is presented in a logical and organized manner. However, there are a few areas where improvements can be made to enhance the clarity and flow of the text. Here are some suggestions:
Sentence restructuring: Consider restructuring a few sentences to improve readability. For example, the first sentence could be revised as: "Attention deficit hyperactivity disorder (ADHD) is a well-known neuropsychiatric disorder in children with an unclear etiology."
Use of hyphens: Ensure proper use of hyphens in compound words. For example, "ADHD-like" should be hyphenated consistently throughout the abstract.
Introduction of the study: Provide a brief introductory statement to introduce the study and its objective. For example, you could start the abstract by briefly mentioning the aim of the study, such as: "This study aimed to investigate the role of miR-200b-3p in ADHD-like symptoms and assess its therapeutic potential."
Clarify the methodology: Include a sentence or two to briefly describe the methodology used in the study. This will provide readers with a better understanding of the experimental approach. For example, you could mention the specific techniques used to measure miR-200b-3p expressions and the administration of high-dose taurine or miR AT.
Use of transition words: Incorporate appropriate transition words to improve the flow between sentences. For example, you could use words like "Additionally," "Moreover," or "Furthermore" to connect related findings or ideas.
Concluding statement: End the abstract with a concise concluding statement summarizing the main findings and their significance. For example, you could state: "These findings suggest that miR-200b-3p may play a role in ADHD-like symptoms and highlight its potential as a therapeutic target for improving these symptoms."
Overall, the abstract provides a clear overview of the study, its objectives, and the main findings. With the suggested improvements, it will become even more effective in conveying the key information to readers.
Reviewer 2 Report
Comments and Suggestions for Authors
The paper of Dr.Chang and colleagues, entitled “Taurine ameliorates ADHD-like symptoms in SHR rats is associated with miR-200b-3p” represents a study on a promising approach for treating ADHD. The combination of taurine and specific micro-RNA (miR-200b-3p) might be a potential therapy, mild and safe. But, although the research topic is very interesting, the manuscript is in immature state. My major points of concern are the following:
- The title does not reflect the contents. The authors have shown mainly changes in the stiatal biochemistry (attenuation of inflammatory cytokines, increased binding to Slit2 protein). The behavioral part, concerning the ADHD symptoms per se, seems to be weak and insufficient. So, the title should be re-phrased to reflect the actual (biochemical) outcomes.
- The abstract is unclear, totally. It is full of abbreviations, which come without any explanation.
- 200Lx illumination is too bright for rats, the observed response should be rather referred to a stress one, not to a spontaneous behaviour.
- the statistical test should be named in the Figure legends
- the statistic should be given in more details: what were taken as variables, categorical factors. Normality of data should be checked before application of ANOVA tests. Why it was one-way ANOVA?
- the results do not look convincing – please, indicate individual values inside the plots, and provide the number of animals under each plot.
- the spontaneous alternation test should not be the only test to say about ADHD symptomps, working memory, etc. Since only the striatum was measured, the authors may speculate rather about the locomotor drive, not the cognitive functions.
- The Discussion is unclear: please, provide more link between the taurine load and non-coding micro-RNA levels. Please, provide a short review paragraph on the animals models of ADHD. The same for the role of striatum/neuroinflammation/neurodegeneration in ADHD-related processes
The language should be improved, sometimes the phrases are paradoxal
Comments on the Quality of English LanguageEnglish must be corrected
Reviewer 3 Report
Comments and Suggestions for Authors
Before this reviewer can proceed to assess the clinical relevance and thus the scientific merit of this manuscript, the authors must compare the dosages they used to human equivalent dosages.
Comments on the Quality of English Language
minor editing suggested
Round 2
Reviewer 1 Report
Comments and Suggestions for Authors
Dear Author,
Thank you for your revised manuscript titled "Both taurine and microRNA–200b–3p antagomir reduce the striatum inflammatory factors and increase the spontaneous alterations in spontaneously hypertensive rats." I appreciate your efforts in addressing the concerns raised in my previous review and making appropriate revisions to the manuscript. I am pleased to see that you have incorporated additional references and included experimental description figures.
Overall, I find that your manuscript has improved significantly, and I believe that the direction of your research holds further value for future investigations. The study of the effects of both taurine and microRNA–200b–3p antagomir on reducing striatum inflammatory factors and increasing spontaneous alterations in spontaneously hypertensive rats is an important area of research with potential implications for understanding and treating hypertension.
I have reviewed the revised manuscript and found the changes to be satisfactory. The addition of relevant references has strengthened the scientific foundation of your study, and the inclusion of experimental description figures provides better clarity and understanding of the methodology employed.
I would like to commend you on your dedication to addressing the comments and suggestions from the previous review. Your revisions have significantly improved the clarity and quality of the manuscript. I am confident that your research will contribute to the existing body of knowledge in this field.
In conclusion, I recommend accepting your manuscript for publication. I believe that your work has met the necessary scientific standards and has the potential to make a valuable contribution to the field. Once again, I appreciate your efforts in addressing the concerns and making appropriate revisions.
Thank you for considering my recommendations, and I look forward to seeing your work published.
Sincerely,
Reviewer
Comments on the Quality of English LanguageThe details of English expression need to be improved
Reviewer 2 Report
Comments and Suggestions for Authors
The manuscript has been accomplished and re-written, as much as it is possible without new experimentation. Although I do not agree completely with the approaches used, now all the things are stated and described clearly - so, the readers will decide.
The abstract provided in the paper is not the same to that provided in the "Response to reviewer". Please, check that all abbreviations are deciphered.
Reviewer 3 Report
Comments and Suggestions for Authors
In lines 397ff, the authors have added the following new text: „The taurine dose used in this study was 45 mmol taurine/kg diet (5.6 g taurine/kg diet), which is equivalent to a dose of 0.9 g taurine/kg diet in humans. The dose of taurine used in this study is much lower than that used for various diseases mentioned above, which provides a rational support for ADHD treatment.”
The authors, however, have failed to give a reference for each and every of their claims. This is inacceptable.
Comments on the Quality of English Languagemoderate editing recommended
